# Moving Beyond Handcrafted Architectures in Self-Supervised Learning

## Abstract

The current literature on self-supervised learning (SSL) focuses on developing learning objectives to train neural networks more effectively on unlabeled data. The typical development process involves taking well-established architectures, e.g., ResNet or ViT demonstrated on ImageNet, and using them to evaluate newly developed objectives on downstream scenarios. While convenient, this neglects the role of architectures which has been shown to be crucial in the supervised learning literature. In this work, we establish extensive empirical evidence showing that a network architecture plays a significant role in contrastive SSL. We conduct a large-scale study with over 100 variants of ResNet and MobileNet architectures and evaluate them across 11 downstream scenarios in the contrastive SSL setting. We show that there is no one network that performs consistently well across the scenarios. Based on this, we propose to learn not only network weights but also architecture topologies in the SSL regime. We show that "self-supervised architectures" outperform popular handcrafted architectures (ResNet18 and MobileNetV2) while performing competitively with the larger and computationally heavy ResNet50 on major image classification benchmarks (ImageNet-1K, iNat2021, and more). Our results suggest that it is time to consider moving beyond handcrafted architectures in contrastive SSL and start thinking about incorporating architecture search into self-supervised learning objectives.

## 1 Introduction

Self-supervised learning (SSL) achieves impressive results on challenging tasks involving image, video, audio, and text. Models pretrained on large unlabeled data perform nearly as good and sometimes even better than their supervised counterparts (Caron et al., 2020; Chen & He, 2021). So far, the focus has been on designing effective *learning objectives* – e.g., pretext tasks (Gidaris et al., 2018; Caron et al., 2018), contrastive (Oord et al., 2018; Chen et al., 2020a) and non-contrastive (Grill et al., 2020) tasks – together with empirical (Cole et al., 2021; Feichtenhofer et al., 2021) and theoretical (Arora et al., 2019; Poole et al., 2019) studies providing key insights and underpinnings. Recent works propose new objectives with a different class of network architectures such as vision transformers (ViT) (Bao et al., 2021) and masked autoencoders (He et al., 2022).

However, there has been little focus on *the role of architectures* in SSL. Currently, the de facto protocol in SSL is to take architectures that perform well on established benchmarks in the supervised setting and to adapt them to the self-supervised setting by plugging in different learning objectives. For example, several existing work on contrastive learning use ResNet (He et al., 2016) as the backbone (Chen et al., 2020a; He et al., 2020). This is partly for convenience. Evaluating different architectures in SSL is computationally expensive; selecting an architecture in advance and fixing it throughout makes it easy to evaluate different learning objectives. This also stems from strong empirical success of those architectures in transfer learning, e.g., CNNs trained on large labeled data provide "unreasonable effectiveness" (Sun et al., 2017; Zhang et al., 2018; Sejnowski, 2020) in a variety of downstream cases. Nonetheless, one implicit assumption is that an architecture that works well in the supervised learning scenario will continue to be effective in the SSL regime.

We argue that this assumption is incorrect and dangerous. It is valid only to a limited extent and the performance starts deteriorating significantly when SSL is conducted on data whose distribution deviates much from the original distribution the architecture was trained on. This is counter to

the promise of SSL, where one can learn optimal representation for a wide range of tasks. One main reason for performance degradation is that different data distributions benefit from different *inductive biases*: An architecture with specific layer types and the wiring between them naturally encodes inductive biases, which may be optimal only for a certain data distribution (e.g., object-centric imagery such as ImageNet) and not for others (e.g., medical and satellite imagery). In fact, numerous studies have shown that standard "recipes" for architecture design do not translate well across different data distributions (Tuggener et al., 2021; Dey et al., 2021; Kolesnikov et al., 2019).

The main objective of this work is to show that *the choice of network architecture crucially matters in SSL*, and that *it is not easy to handcraft architectures that are effective across different SSL scenarios*. To see this, recall that the goal of SSL is to learn data representations capturing important features and attributes that generalize well across various downstream tasks. There has been extensive literature on the expressivity of neural networks (Raghu et al. (2017); Zhang et al. (2021a) and references therein); one of the important conclusions is that the network topology plays a significant role in determining the *expressivity*, i.e., the kinds of functions a network can approximate is bounded by the network capacity and available sample size in the finite sample regime. This implies that, in practice, SSL with a fixed architecture learns representations only within the scope of function space induced by the *pre-selected* architecture topology, and therefore, the ultimate success of SSL can be achieved when it finds optimal architecture from certain search space in conjunction with its weights for specific data distributions.

In this paper, we establish extensive empirical evidence showing that *architecture matters in self-supervised learning*. We do this in two sets of large-scale studies. First, we sample 116 variants of ResNet (He et al., 2016) and MobileNet (Sandler et al., 2018) architectures with different topologies and evaluate them on 11 downstream tasks in the SSL setting. We pretrain all models under the same setting, optimizing the SimCLR objective (Chen et al., 2020a) on ImageNet (Deng et al., 2009), and investigate if there exist any correlation between these models in downstream performance on different datasets. We observe no strong correlation, except for tasks highly similar to ImageNet. We further show that ImageNet downstream performance, the gold standard benchmark in the SSL literature, is not indicative of performance on other downstream tasks. This implies that we need to be careful in choosing an architecture for evaluating any newly developed SSL objectives, as one might get different conclusions based on different network architectures.

This subsequently raises the question: *Can we improve SSL by learning not only network weights but also architectures directly optimized for the given dataset?* It removes the burden of manually searching for effective architectures in SSL, and if we succeed, it can substantially improve performance of SSL. To test this hypothesis, as the second set of our study, we apply a well-established NAS algorithm (Cai et al., 2018) to the SSL setting. Unlike the typical NAS setting that optimize on a *labeled* target dataset, we search for optimal architectures directly on an *unlabeled* pretraining dataset via contrastive learning (Chen et al., 2020a). We evaluate our "NAS + SSL" framework on datasets with different distributions, ImageNet-1K and iNat 2021 (Van Horn et al., 2021), and show that self-supervised architectures consistently outperform handcrafted ones in the same parameter range (MobileNetV2 and ResNet18) across 11 downstream tasks. This provides strong evidence suggesting the importance of learning architecture topologies in addition to their weights in SSL.

Our work focuses on studying the role of architectures in contrastive SSL with the SimCLR framework for CNN-based architectures such as ResNets and MobileNets. As a first step in this direction, we provide an in-depth analysis through large-scale experiments in this specific (yet limited) setting. Extending our study to different SSL approaches (He et al., 2020; Grill et al., 2020; He et al., 2022) and architectures such as ViTs would be an interesting direction but beyond the scope of this paper.

In summary, our main contributions are: **1)** We establish extensive evidence showing that there isn't one network architecture performing consistently well across different downstream scenarios in the SimCLR setting. We show this using 116 variants of ResNet and MobileNet architectures pretrained on ImageNet and evaluated on 11 downstream datasets. **2)** We show that ImageNet performance (the gold standard in SSL benchmark) is not always indicative of downstream performance. This means that findings about SSL objectives shown only on ImageNet do not generalize across other data distributions. **3)** We propose to self-supervise a CNN architecture topology and its network weights on unlabeled data. We show that self-supervised architectures outperform handcrafted ones in a similar parameter range for the SimCLR setting across different downstream datasets.

## 2 RELATED WORK

**Role of architectures in SSL.** There has been significant progress in learning representations via SSL (Gidaris et al., 2018; Chen et al., 2020a;c; Caron et al., 2020). Ericsson et al. (2021) provide an overview of different SSL setups and their performance on downstream tasks. Most works focus on improving self-supervised objectives to develop better representations while keeping architectures fixed. Our focus is orthogonal to this line of work. We study the role of architectures in SSL and investigate the benefits of self-supervising architecture topologies along with network weights.

Similar to ours, Kornblith et al. (2019) analyze transfer performance of different architectures pretrained on ImageNet across several downstream datasets. However, they focused on supervised learning, which need not translate to self-supervised setups (Kolesnikov et al., 2019). Caron et al. (2021) propose a self-distillation based SSL objective and compare the performance of ResNets with ViTs. In contrast, we focuse on contrastive SSL and the importance of architecture topologies for the class of CNNs. Kolesnikov et al. (2019) is the most related to ours but are limited to pretext-task based SSL and show results only for a few variants of VGG and ResNet. In comparison, we conduct a study on a much larger scale in a contrastive learning setup. Also, we provide insights into generalization performance of the networks across different datasets. Crucially, unlike all previous work in SSL, we propose to learn both architecture topologies and their network weights.

**NAS and SSL.** The literature on NAS has rapidly progressed in the past years; we refer the reader to the NAS survey by Elsken et al. (2019). Here we focus on the most directly relevant work to SSL. Mellor et al. (2021) search for networks without any training and verify their effectiveness of supervised benchmarks. Liu et al. (2020) show that highly performant architectures can be found without using any supervised labels during the search itself, and propose using DARTS (Liu et al., 2019) with self-supervised proxy tasks to search for architectures which will perform well on supervised datasets. In a similar vein, Zhang et al. (2021b) use random labels during the search phase of NAS, and Yan et al. (2020) learn representations of architectures via SSL and use them to improve NAS. Li et al. (2021) introduce a new self-supervised training scheme to search for architectures in a new hybrid search space. One common theme in this line of work is that they aim to *harness SSL in aid of NAS*. In contrast, we aim to utilize NAS to hunt for architectures which will perform well for self-supervised learning, thereby *harnessing NAS in aid of SSL*.

## 3 DOES ONE NETWORK RULE THEM ALL IN SSL?

We conduct a large scale study to investigate whether a particular architecture topology can be consistently effective across a wide range of SSL scenarios. To this end, we sample 116 models with different architecture topologies and analyze their performance on 11 downstream tasks.

To maximize generality while taming complexity of our study, we choose two most representative CNN architectures: ResNets and MobileNets. The former is the de facto backbone for numerous modern visual models (Kirillov et al., 2019; Wu et al., 2019) and SSL approaches (Caron et al., 2020; Chen & He, 2021), while the latter is used in low-resource setups (Cheng et al., 2017). We create 69 ResNet-like and 47 MobileNet-like architectures for our evaluation, varying the number of blocks at each of the 4 stages of a ResNet, the block structure (BasicBlock and Bottleneck), width, and number of groups. For MobileNet, we vary the width multiplier parameter and the number of blocks in each of the 6 stages. We choose the SimCLR objective (Chen et al., 2020a) for our experiments and pretrain each of the 116 architectures on ImageNet-1K. Owing to the large scale nature of the experiments and computational constraints, we limit the pretraining to 100 epochs and a batch size of 512. Each of these jobs takes roughly 1 day to finish on a system with $8\times$ V100 (32GB) GPUs. We report linear evaluation results in all cases.

**ImageNet performance is not indicative of downstream performance for SSL.** To examine the correlation between ImageNet vs. downstream performance, we compute the Spearman's rank correlation coefficient $\rho$ on top-1 validation accuracy between every dataset pair, shown in Fig. 1. We also show scatter plots in Fig. 2 revealing the relationship between ImageNet vs. downstream performance on the most representative cases; the complete set of scatter plots are in the appendix.

For ResNet-like architectures (Fig. 2 top row), we see strong correlation between ImageNet and CIFAR-100 ($\rho = .8$) as these datasets contain similar categories; about 90 classes in CIFAR-100 are

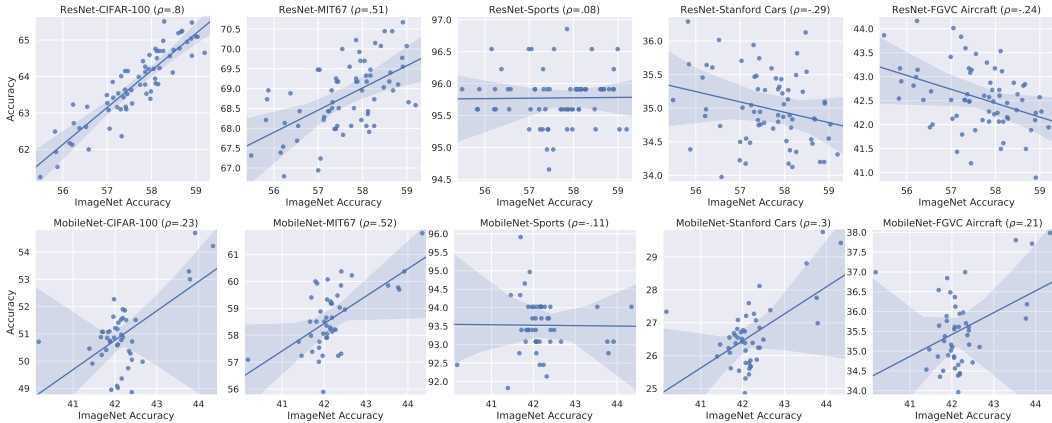

Figure 2: **ImageNet performance is not indicative of downstream performance in SSL.** We show linear evaluation top-1 accuracy of ImageNet (x-axis) vs. downstream datasets (y-axis) obtained from variations of ResNet (top) and MobileNet (bottom); all models are pretrained on ImageNet-1K using SimCLR under the same protocol. The solid lines and shaded areas indicate fitted regression models and their confidence intervals.

the same class or a superclass in ImageNet. A similar observation is made (in Fig. 1) for Stanford Dogs ($\rho = .77$) due to the 120 dog categories in ImageNet. However, we observe high variance in transfer performance for out-of-domain datasets, e.g., Flowers ($\rho = .35$), represented by only two ImageNet categories (daisy and yellow lady slipper). Correlation becomes negative on Stanford Cars ($\rho = -.29$) and FGVC Aircraft ($\rho = -.24$), likely because ImageNet contains only a few categories of cars (10 classes) and aircraft (4 classes). Low correlation means network ranks are inconsistent and models performing well on ImageNet do not keep their precedence in other tasks.

For MobileNet-like architectures (Fig. 2 bottom row), we see overall much lower correlation with ImageNet performance; the ResNet space showed correlation at least for in-distribution datasets. For e.g., we see correlation between ImageNet and CIFAR-100 drops from $\rho = .8$ (ResNet) to $\rho = .23$ (MobileNet). For other datasets like MIT67, correlation is higher ($\rho = .52$) but still less meaningful due to high variability in performance (notice the cluster around 42% accuracy). These results indicate that MobileNets are even less tuned towards ImageNet than ResNets and any handcrafted architectures in this space is likely to be suboptimal on ImageNet and other downstream datasets.

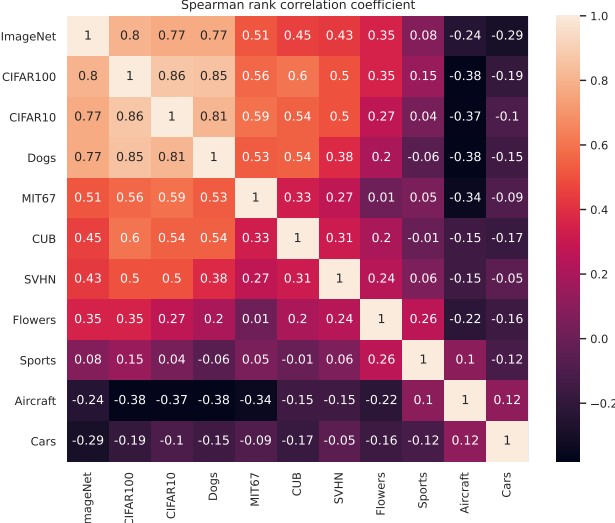

Figure 1: **ImageNet performance isn't indicative of downstream performance in SSL.** We show rank correlation between each pair of top-1 accuracy on 11 downstream tasks obtained from ImageNet-pretrained ResNets. We see no strong correlation except for ones similar to ImageNet, e.g., CIFAR-10/100 and Dogs120.

Our results highlight that the same architecture recipes in terms of ImageNet accuracy, which is frequently used as a predictor for various self-supervised tasks, do not work well for different datasets in the SimCLR setting.

**Larger networks do not always perform better in contrastive SSL.** In general, large-parameter models yield better performance on ImageNet, both in supervised (Kornblith et al., 2019) and SSL setups (Chen et al., 2020a). We examine whether this trend holds for downstream datasets some of which are widely different from ImageNet. We follow the same setup as above and compute the

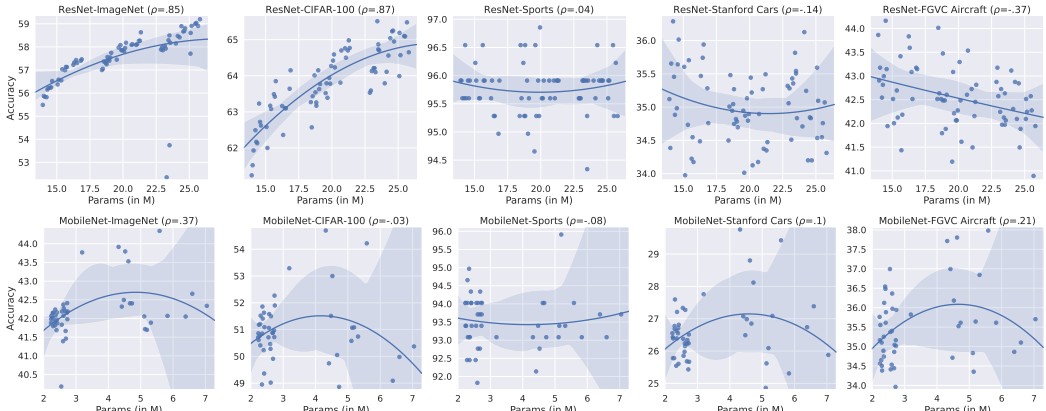

Figure 3: **Larger models do not always perform better in SSL.** We show the model size in terms of parameter counts (x-axis) vs. top-1 accuracy on different datasets obtained from variations of ResNet (top) and MobileNet (bottom) architectures; all models are pretrained on ImageNet-1K using SimCLR under the same protocol.

correlation between number of model parameters and top-1 validation accuracy on different datasets. Fig. 3 shows scatter plots of the most representative results; full results are in the appendix.

For ResNet-like architectures (Fig. 3 top row), we do not see strong trends indicating larger models always perform better. In fact, on some downstream tasks we see negative correlation (Aircraft, $\rho = -.37$) with lighter networks being favored for better performance, or even non-linear relationship, e.g., notice the slight "U" pattern on Stanford Cars ($\rho = -.14$), indicating the behavior of ResNets on these datasets are wildly unexpected. Unsurprisingly, there is strong correlation with ImageNet ($\rho = .85$) and CIFAR-100 ($\rho = .87$), likely because ResNets are heavily hand-tuned on the kinds of images observed in these datasets. These results clearly suggest that the same architecture recipes which work well for ImageNet (increasing parameters through depth and width) do not hold for other downstream scenarios.

For MobileNet-like architectures (Fig. 3 bottom row), we see that there exists almost no interpretable trend. The fitted regression models (solid lines) and their confidence intervals (shaded area) show that the relationships are highly non-linear and non-monotonic; we shouldn't read too much into the correlation coefficients (reported for completeness). These results indicate that MobileNets do not favor any one architecture and it is heavily reliant on the dataset it is trained on.

Our results suggest that larger models are not always better in SSL; lighter models can outperform heavier models on tasks like FGVC Aircraft. Previous work (Chen et al., 2020b) showed that increasing network depth/width improves downstream performance. However, they vary depth at a coarser level with 50/100/150 layers and width with $1 \times / 2 \times$, leading to a large swing in network parameters (24-795M); here we show the same is not true at a finer level. Our results provide evidence that ResNet architectures are tuned to scale well on ImageNet but not the others; the trend is even weaker for MobileNet-like architectures, suggesting they are optimized for computational efficiency and not for achieving high accuracy on any particular dataset.

**There is no winner in the battle of top vs. bottom heavy networks in SSL.** Raghu et al. (2017) showed that CNNs are more sensitive to lower (initial) layer weights, suggesting that "not all weights are created equal" across layers. This raises the question: If CNNs are more sensitive to lower layers, will increasing the parameter count for lower layers yield better performance in SSL? To get insights into this, we split a network into two halves: "top" (layers closer to output) and "bottom" (closer to input). We use the ratio of top to bottom parameters as a measure of networks being "top-heavy" (high ratio) or "bottom-heavy" (low ratio).

Fig. 4 shows the downstream accuracy for ImageNet, Stanford Cars and Sports against this ratio. From the top-left subplot, we see that ResNet-ImageNet accuracy tend to increase with high top:bottom ratio, showing top-heavy networks generally perform better than the bottom-heavy counterparts. However, we no longer observe such trend in other datasets, and with MobileNets (Fig. 4 bottom row) we do not see such trend even for ImageNet.

An important point to realize: **It is necessary to allocate the right portion of parameters to different layers of a given network topology**, instead of allocating parameters in the top-heavy or

bottom-heavy fashion assuming one will generally lead to better performance. ResNets and many other handcrafted CNNs (Simonyan & Zisserman, 2014; Szegedy et al., 2016; Huang et al., 2017) are usually top-heavy because of GPU memory limits; bottom-heavy networks occupy more memory in terms of activation maps. MobileNets alleviate this to some extent with lighter convolutions, allowing it to have more parameters in early layers. In object detection, Liang et al. (2019) also show that allocation of computational resources in the backbone is important for improved performance. While this architectural difference provides explanations about the wildly different trends we observe above, the key message here is that one recipe (top vs. bottom heavy) does not apply equally to different architectures, bolstering our claim that one network doesn't rule them all in SSL.

**Key takeaway: We need to move beyond hand-crafted architectures in SSL.** The three main observations above imply that finding an optimal architecture could be an important missing piece for self-supervised learning. Our results show that the current practice in designing SSL objectives – i.e., optimizing for ImageNet performance based on ResNet backbones – could lead to misleading conclusions which do not generalize to other downstream scenarios. Also, the general belief that "the larger the better" in model size do not really hold in SSL, e.g., smaller

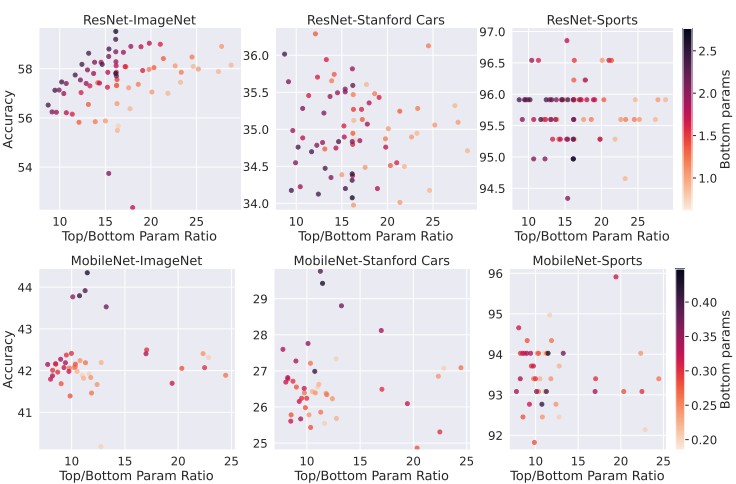

Figure 4: **There is no winner in the top vs. bottom battle in SSL.** Except for ResNet-ImageNet (top-left), we see no strong trend that suggests either top-heavy or bottom-heavy networks perform better.

ResNets can outperform larger ones even if they are pretrained following the same protocol. Let's say, based on these observations, one is compelled to hunt for a new architecture geared specifically towards SSL. Our top vs. bottom analysis suggests that it can be extremely tricky to find the right architecture topology with optimal parameter allocation across different layers. All this suggests that it is time to consider moving beyond handcrafted architectures in SSL and start thinking about *searching for optimal architectures* as part of self-supervised learning objectives.

## 4 NEURAL ARCHITECTURE SEARCH FOR SSL

We turn to the idea of learning both the architecture topology and its network weights in an SSL framework, using NAS to improve SSL (rather than using SSL to improve NAS). It is important to draw a clear distinction between our idea and prior work that used SSL to improve NAS (Li et al., 2021; Liu et al., 2020; 2019; Yan et al., 2020; Zhang et al., 2021b) as well as work that used NAS to improve supervised learning (Elsken et al., 2019); our goal here is to show the benefit of harnessing NAS in aid of SSL and not for comparison with more recent SOTA NAS approaches.

While examining this idea appears to be straightforward, it requires careful design of experiments. The biggest hurdle is that both NAS and SSL require heavy compute resources; the former needs a search space large enough to cover a comprehensive range of architecture topologies, while the latter requires large datasets and batch sizes to be effective. This calls for an efficient framework to conduct our study. Furthermore, we need datasets large enough to pretrain the models on, and different enough to investigate the importance of data-dependent architectures in SSL. To meet our desiderata, we choose ProxylessNAS (Cai et al., 2018) as our NAS algorithm, MobileNet as our search space, and ImageNet-1K and iNat2021 (Van Horn et al., 2021) as our pretraining datasets.

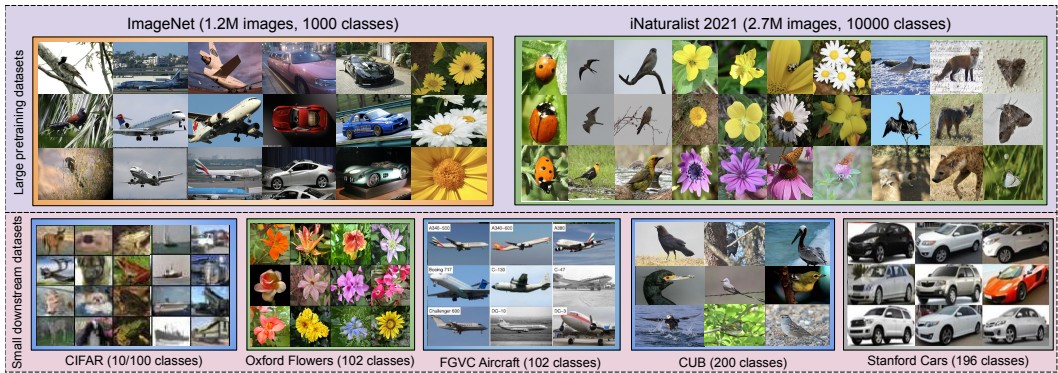

Figure 5: **Samples from datasets used in our study.** We choose these datasets because of the apparent domain shift across them. ImageNet contains general yet coarsely categorized images compared to iNat2021, which contains an order of magnitude higher number of fine-grained categories; although some images look similar to each other, every image shown belongs to a different category highlighting the fine-grained nature of this dataset. The downstream datasets, except for CIFAR, are similarly fine-grained but on different domains.

## 4.1 EXPERIMENTAL SETUP

**SSL objective.** We use SimCLR (Chen et al., 2020a), one of the most well-established contrastive SSL framework. We follow the same augmentation methods and hyperparameter settings as in Chen et al. (2020a). While more recent SSL works exist (Caron et al., 2020; Chen & He, 2021), they are similar to SimCLR by utilizing a contrastive learning based objective. We adopt SimCLR for its simplicity and leave analysis of other SSL approaches for future work.

**NAS algorithm.** We choose ProxylessNAS for two reasons: efficiency and flexibility. One-shot NAS algorithms produce an architecture topology in a two-step process. They first find an optimal cell structure by solving a proxy task over a small dataset (e.g., CIFAR-10) and a smaller architecture (e.g., 8 cells), and then stack/repeat the best found cell topology for the target task (e.g., ImageNet with 20 cells). This reduces complexity at the cost of flexibility and introduces an optimization gap (Chen et al., 2021), requiring strong correlation between proxy and the actual target datasets. In contrast, ProxylessNAS produces an architecture by directly optimizing on a target task, as it can significantly ameliorate memory requirements of one-shot NAS methods. To ensure flexibility, it uses a "supernet" with a broad range of candidate operations orchestrating depth (via zero operations), width (via wider convolutions), and block structure (by allowing for operations to differ by level). At each training step, it optimizes one "subnet" on the target task as a surrogate, which greatly reduces compute and memory requirements.

**Datasets.** We use ImageNet-1K and iNat2021 to pretrain our models and evaluate them on their validation sets as well as on 10 downstream datasets used in Section 3. We deliberately choose the two pretraining datasets as they exhibit widely different characteristics, i.e., ImageNet-1K contains a variety of objects and scenes, while iNat2021 contains fine-grained species covering the tree of life. The former contains many inorganic object categories not present in the latter. This creates a domain gap, which allows us to investigate the importance of data-dependent architectures in SSL.

iNat2021 contains 2.7 million images (twice the size of ImageNet) representing 10K species (ten times more than ImageNet). To investigate the effect of dataset size during architecture search and pretraining, we use both the full and the mini versions of iNat2021 – the latter contains 500K images representing the same 10K classes. This gives us pretraining datasets at three different scales: 500K (iNat2021-mini), 1.2M (ImageNet), 2.7M (iNat2021).

**Implementation details.** For ProxylessNAS, we replace the original supervised classification loss with the contrastive loss of SimCLR and remove the latency loss as we currently do not consider hardware-constrained scenarios. We follow the original training schedule, i.e., a warmup phase for 40 epochs, which optimizes only the network weights and not the NAS parameters, followed by a search phase for 120 epochs. We use the SGD optimizer for network weights and the ADAM optimizer for NAS parameters, using initial learning rates of 0.25 and 0.1, respectively, and use the cosine decay schedule for both.

To make our experiments tractable, we use the MobileNetV2 search space which typically yields 3 to 18 million parameters; the ResNet search space is larger, yielding 20 to 60 million parameters. Working with smaller models means we can use large batch sizes, which is important for contrastive learning to work effectively; we use the batch size 640 given our computational budget. The candidate set of NAS operations consists of mobile inverted bottleneck convolution (MBConv) with kernel sizes $\{3, 5, 7\}$, expansion ratios $\{3, 6\}$ and zero operations. A higher expansion ratio enables a wider network with more channels for convolutions, while zero operations allow for choosing to remove operations, thereby learning the optimal depth. Once the search is done, we take the architecture and discard the learned weights; we train it again from scratch using the SimCLR objective on different datasets. This allows us to compare different architectures on fair ground. After pretraining, we conduct linear evaluation on all downstream datasets.

## 4.2 Results and Discussion

**Self-supervised architectures outperform handcrafted architectures in SSL.** Table 1 compares our self-supervised architectures to MobileNetV2 and ResNet18/50, by searching, pretraining, and evaluating on ImageNet-1K, iNat2021 and iNat2021-mini. We report linear evaluation results on validation splits. The results show that our self-supervised architectures outperform MobileNetV2 by a large margin, even with similar parameters (about 3M).

Table 1: **Searched architectures vs. handcrafted architecture results.** We search, pretrain, and evaluate ours on each of the three datasets in the last three columns. [†]SOTA results (in gray) from Chen et al. (2020a) for ImageNet and Cole et al. (2021) for iNat21 require larger batch sizes and longer training.

| Model | Params | Batch | Epochs | ImageNet | iNat21 | iNat21-mini |
|---|---|---|---|---|---|---|
| MobileV2 | 3.5M | 640 | 100 | 41.9 | 30.2 | 13.6 |
| Ours | 3.3M | 640 | 100 | **55.3** | **40.3** | **14.7** |
| ResNet18 | 11M | 640 | 100 | 49.8 | 30.3 | 20.1 |
| ResNet50 | 23.5M | 640 | 100 | 58.9 | 41.3 | 23.4 |
| Ours | 12-18M | 640 | 100 | **59.1** | **43.8** | **25.1** |
| ResNet50 | 23.5M | 4096 | 1000 | 69.3[†] | 50.6[†] | - |

The self-supervised architectures also beat ResNet18 and ResNet50, with even smaller model size than ResNet50. The superior downstream performance of our approach should not be attributed solely to NAS, as the architecture search was performed without ever solving the downstream tasks. It is rather the incorporation of NAS into SSL that improved the quality of representations, leading to downstream performance boost. This shows the effectiveness of learning both the architecture topology and its weights in SSL.

**Do self-supervised architectures generalize well to different data distributions?** We take the three architectures searched on each dataset, discard their learned weights, and pretrain them on each dataset, yielding 9 pretrained models. We then evaluate the performance directly on validation splits of the respective datasets. Table 2 shows that transferring an architecture from iNat2021 to ImageNet leads to a marginal

Table 2: **Self-supervised architecture transfer results.** We evaluate architectures in the cross-dataset setting, pretraining and evaluating the searched architectures across three datasets (last three columns).

| Searched on | Params | ImageNet | iNat21 | iNat21-mini |
|---|---|---|---|---|
| ImageNet | 12-18M | **59.1** | 21.5 | 23.9 |
| iNat21 | 12-18M | 58.3 | **43.8** | **27.9** |
| iNat21-mini | 12-18M | 58.0 | 22.4 | 25.1 |

performance drop compared to an architecture optimized directly on ImageNet (59.1% to 58.3%); both these architectures still outperform handcrafted ResNet18 (49.8%) with a comparable model size. However, transferring an architecture from ImageNet to iNat2021 deteriorates performance significantly (43.8% to 21.5%). This implies an interesting finding, i.e., iNat21-searched architectures seem to be more resilient to domain shift than ImageNet-searched architectures. This could be due to the difference in dataset size (iNat21 has twice as many images as ImageNet), or due to the fine-grained nature of iNat21 resulting in an overall more difficult instance discrimination task (Chen et al., 2020a) that leads to more discriminative representations. The effect of dataset size on architecture search is also shown on iNat21-mini results. While transferring an architecture from ImageNet to iNat21-mini shows an expected drop in accuracy (25.1% to 23.9%), transferring from the larger iNat21 improves performance (25.1% to 27.9%). As both datasets are in the same domain and only differ in number of samples per class, higher search dataset size is the driving factor behind the gains in accuracy while pretraining on a smaller version of the dataset.

**Downstream transfer experiments.** The results above show that self-supervised architectures are superior to handcrafted architectures when tested in in-distribution settings, which might reflect a practical use case of self-supervised pretraining in the real-world setting (e.g., one has access to only

Table 3: **Downstream transfer results.** We categorize downstream datasets as in-distribution (green) and out-of-distribution (red) relative to the pretraining dataset based on class overlap; best viewed in color. We see that self-supervised architectures generally perform better on in-distribution downstream scenarios.

| Pretrain Dataset | Arch. | Params | Pretrain Val. Set | CUB | NABirds | CIFAR10 | Oxford Flowers | Stanford Dogs | Food101 | Sport | Stanford Cars | MIT67 | FGVC Aircraft |
|---|---|---|---|---|---|---|---|---|---|---|---|---|---|
| ImNet | MobileV2 | 3.5M | 41.9 | 20.1 | 14.2 | 75.8 | 85.4 | 35.7 | 51.6 | 94.0 | 26.4 | 57.5 | 35.8 |
| | Ours | 3.3M | **55.3** | **31.8** | **24.4** | **78.8** | **91.7** | **49.2** | **61.7** | **94.3** | **30.4** | **62.5** | **38.1** |
| | ResNet18 | 11M | 49.8 | 27.0 | 19.0 | 79.9 | 89.9 | 44.1 | 55.8 | 94.6 | 27.8 | 62.1 | 36.7 |
| | ResNet50 | 23.5M | 58.9 | 31.4 | 24.6 | **85.9** | **92.8** | **52.3** | **65.7** | 94.3 | **35.1** | **69.6** | **42.0** |
| | Ours | 3-18M | **59.1** | **34.3** | **26.1** | 81.8 | 92.2 | 51.0 | 64.7 | **94.7** | 33.2 | 66.1 | 39.4 |
| iNat21 | ResNet18 | 11M | 30.3 | 26.1 | 19.0 | 73.2 | 92.8 | 31.3 | 55.3 | 92.1 | 18.9 | 49.9 | 32.7 |
| | ResNet50 | 23.5M | 41.3 | 31.2 | 23.2 | 75.1 | **95.1** | **39.7** | **65.1** | **94.3** | **22.3** | **55.0** | **37.6** |
| | Ours | 3-18M | **43.8** | **32.7** | **24.1** | **76.1** | 94.7 | 39.0 | 63.1 | 93.1 | 20.1 | 49.0 | 34.9 |

small labeled but large unlabeled data from the same distribution). We now evaluate our approach on a downstream transfer scenario with possible domain shift and with much smaller datasets. To this end, we again use the 10 downstream tasks used in Section 3, which contain datasets coming from both in-distributions and out-of-distributions relative to the pretraining datasets.

Table 3 summarizes the results (we color code in/out-of-distribution datasets based on our crude categorization; see appendix for our justification). We first compare our self-supervised architectures to MobileNetV2 in the same parameter range (3.5M vs 3.3M; top two rows). We notice that self-supervised architectures significantly outperform MobileNetV2 in all datasets *regardless of* distributional shift. This is encouraging (i.e., self-supervised architectures can learn generalizable representations) but at the same time not totally surprising (i.e., MobileNet is optimized for efficiency and not for accuracy). Next, we compare ours to ResNet18 that has a similar parameter range although belonging to a class of architectures much different from our search space. Ours outperforms ResNet18 on all pretraining and evaluation datasets by a considerable margin. This shows that our approach is generalizable and can outperform architectures in the ResNet18 search space even though they are generally more computationally expensive than the MobileNet search space.

Finally, we compare ours to ResNet50 which is computationally heavier compared to ResNet18. We preface our analysis with a caveat that our self-supervised architectures are almost half the capacity of ResNet50, limiting their representational power. Keeping this in mind, we see that our approach starts to fail in some of the in-distribution and all of the out-of-distribution scenarios (red shaded cells). This is somewhat disappointing but perhaps expected: self-supervised architectures naturally encode *inductive biases* specific to the dataset they were optimized on. When a distributional shift happens, their performance can start deteriorating because out-of-domain data might require a different set of inductive biases. The strong performance by ResNet50 imply that the model might be striking the right balance across those datasets in terms of inductive biases, but our results in Table 1 and 2 show that ResNet50 can be less effective on newly developed datasets such as iNat2021.

## 5 CONCLUSION

This work lays the ground for moving beyond handcrafted architectures in SSL. By conducting large-scale experiments with 116 architectures and 11 downstream tasks, we established extensive empirical evidence showing that there isn't one architecture that performs consistently well across different downstream scenarios in SSL. Motivated by this, we proposed to move beyond handcrafted architectures and learn both an architecture topology and its network weights in SSL. We provided convincing results demonstrating that the self-supervised architectures significantly outperform handcrafted MobileNetV2 and ResNet18 architectures on 11 downstream tasks, and competitively with ResNet50 even with almost half the model size. We re-emphasize that improvements are not solely due to NAS, as the architecture search was performed by solving SSL and not by optimizing directly on downstream tasks as in the typical NAS setting.

Our work barely scratches the surface and opens up many doors for future directions. Will our findings hold for different architectures such as Transformers and different modalities such as video and text? How can we make architecture search more effective for SSL? Can ideas from domain generalization improve the transferability of self-supervised architectures in the out-of-distribution setting? Or is it even the right idea to expect learned architectures to generalize to widely different domains? We hope the readers are as excited as us to investigate these challenging questions.

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

APPENDIX

Fig. 6 provides an overview of our work. We show that no single handcrafted architecture performs consistently well across different tasks. It is therefore imperative to optimize for architecture topologies along with network weights for a specific task. Through extensive empirical results we show that such self-supervised architectures outperform their handcrafted counterparts in the same search space on the respective tasks.

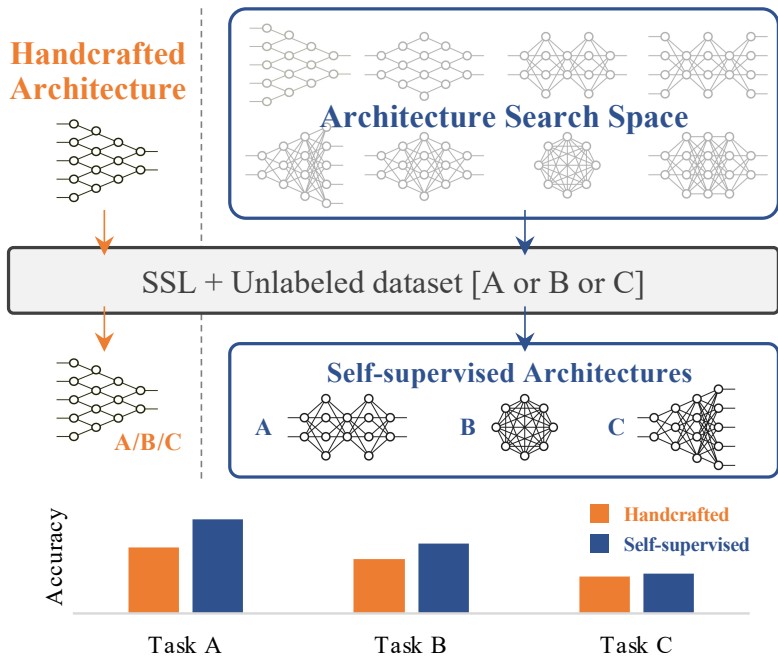

Figure 6: Conventional SSL frameworks learn network weights for a fixed handcrafted architecture (left). We show that learning architecture topologies along with their weights can improve performance in SSL (right).

## A    SUPERVISED TRAINING PERFORMANCE OF SELF-SUPERVISED ARCHITECTURES

In addition to evaluating the performance of the searched architectures for SSL, we analyze their supervised training performance. We use the searched architectures and directly train them, from scratch without any pretraining, on downstream datasets using the supervised labels. Results are summarized in Table 4. We include architectures searched on ImageNet and iNat21, and MobileNetV2 for reference. It shows the searched architectures perform well even in the supervised setting, outperforming the handcrafted MobileNetV2 on most of the downstream datasets. However, a performance degradation is observed in out-of-distribution datasets like Stanford Cars and FGVC Aircraft. This is in line with the discussion in Section 4.2 of the main paper where the searched architecture performances deteriorate with distributional shift. Nevertheless, for the more in-distribution datasets, we obtain higher accuracies showing that the searched architectures are suitable for supervised training as well.

Table 4: **Supervised performance of searched architectures**.

|              | CUB      | CIFAR10  | CIFAR100 | Food     | Flowers  | Sport    | Cars     | Aircraft |
|--------------|----------|----------|----------|----------|----------|----------|----------|----------|
| MobileNetV2  | 58.2     | 93.7     | 71.9     | 80.8     | 90.0     | 94.2     | **88.6** | **81.4** |
| Ours (ImNet) | 57.9     | 93.1     | 74.7     | 78.1     | 96.7     | 95.0     | 71.0     | 75.5     |
| Ours (iNat21)| **64.5** | **93.9** | **75.8** | **81.6** | **98.1** | **96.3** | 72.3     | 72.9     |

## B    CLASS MAPPING FROM IMAGENET TO DOWNSTREAM DATASETS

We provide a justification for characterizing downstream datasets as in-distribution/out-of-distribution with respect to ImageNet as shown in Table 3 of the main paper. We provide a rough class mapping between ImageNet and the 10 downstream datasets. Note that obtaining an exact class mapping is difficult due to only an approximate mapping existing between any 2 datasets. In addition, there can be classes which contribute to improved features for another class while still being semantically different. For example, zebra (n02391049) can contribute to improved features for horses (sorrel-n02389026) due to similar shapes. We now list datasets with corresponding ImageNet classes/superclasses. While some superclasses can contain additional subclasses in the WordNet hierarchy, we restrict to only those classes in the ImageNet-1k dataset. Numbers in bracket denote the total number of classes roughly overlapping.

- CIFAR-10 (270):  vehicle (n4524313), bird (n1503061), feline (n2120997), frog (n1639765), dog (n2084071), sorrel (n2389026)
- Stanford Dogs (120): dog (n2084071)
- CUB (60): bird (n1503061)
- NABirds (60): bird (n1503061)
- Food101 (20): nutriment (n7570720), beverage (n7881800), foodstuff (n7566340), sandwich (n7695965), bagel (n7693725), guacamole (n7583066), chocolate sauce (n7836838), carbonara (n7831146), french loaf (n7684084), pretzel (n7695742)
- Stanford Cars (10): car (n2958343)
- FGVC Aircraft (3): airliner (n2690373), warplane (n4552348), airship (n2692877)
- Oxford Flowers (2): yellow lady's slipper (n12057211), daisy (n11939491)
- MIT67 (0): -
- Sports(0): -

Due to the inductive biases encoded during the search process specific to the dataset it is searched on, the self-supervised architecture performs well on more in-distribution datasets like CUB or NABirds. However, we see that for datasets like Stanford Cars and subsequent ones, there is little direct class overlap with ImageNet classes. This leads to lesser images being available for self-supervised pretraining which are in-distribution for these datasets. Consequently, due to the relatively out-of-distribution nature of these datasets we see in Table 3 of the main paper, our self-supervised architectures are outperformed by the ResNet-50 baseline.

## C    ADDITIONAL IMPLEMENTATION DETAILS

We sample ResNet architectures by varying the number of blocks at each of the 4 stages choosing from the set of $2, 3, 4$ blocks and choose the ones in the parameter range shown in Fig. 3 while also fitting in GPU memory. For MobileNets, we have 7 sequences (stages) and a higher variation of the number of blocks from [2-6] while also choosing the width parameter from the set $1.0, 1.2, 1.4, 1.6, 1.8, 2.0$ and choose the ones in the 2M-7M parameter range and fitting in GPU memory. Note that a high number of blocks in the earlier stages take significantly more GPU memory due to larger feature map sizes.

For the architecture search phase, we use the optimizer hyperparameters as explained in Sec. 4.1 of the main paper. We use a weight decay of $4e^{-5}$ for the weight parameters excluding batch normalization parameters. The initial convolution is a 3x3 convolution with stride 2. The network consists of 6 stages with 4 cells in the first 5 stages and 1 cell in the last stage. By default, the number of channels at each stage is 24, 40, 80, 96, 192, 320, which is multiplied by a constant width multiplier. We downsample it by a factor of 2 at the beginning of the first, second, third and fifth stage. Other architecture details are the default ones used in Cai et al. (2018). We use the same projection head as used normally for SimCLR Chen et al. (2020a) on top of the backbone network, which is a 2048 dimensional hidden layer and 128 dimensional output layer. For evaluation, we remove the projection head and use the output of the network backbone as the feature extractor. Augmentations are the same as in SimCLR with random resize scaling and cropping, flipping and color jitter. A temperature value of $\tau = 0.1$ is set for the contrastive loss.

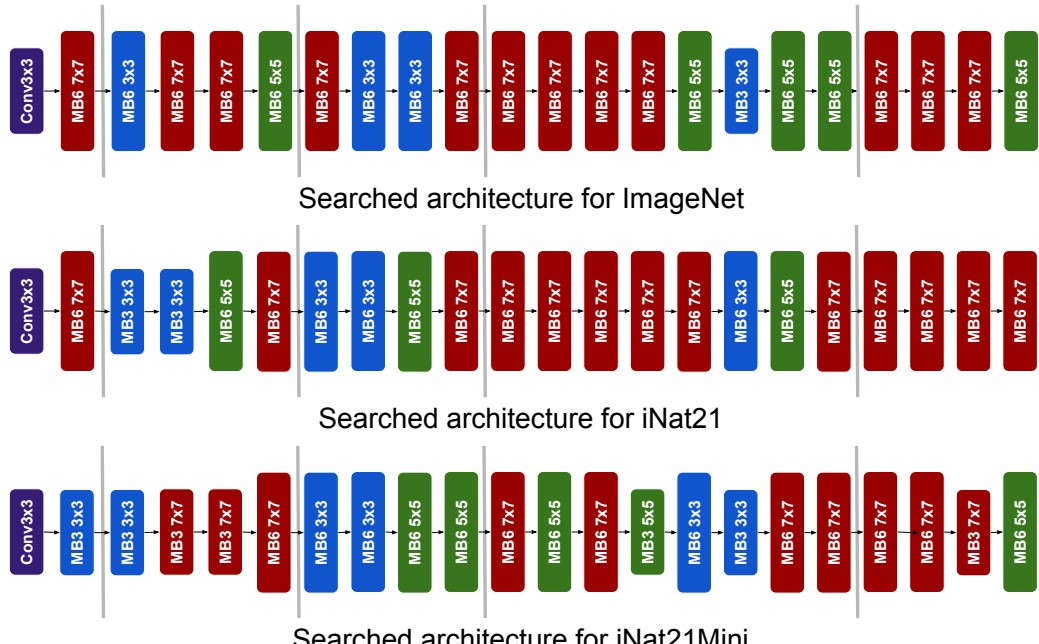

Figure 7: **Self-supervised architectures for different pretraining datasets.** MB3 and MB6 are the mobile inverted convolutions with expansion ratio of 3 and 6 respectively. We see that the majority of the preferred convolutions is MB6 $7 \times 7$ suggesting that the network prefers convolutions with more parameters for the self-supervised regime due to lots of data. For smaller datasets like iNat21Mini, MB6 convolutions are not as strongly preferred.

## D    VISUALIZING SELF-SUPERVISED ARCHITECTURES FOR DIFFERENT PRETRAINING DATASETS

We visualize the types of convolutions searched at a 1.75x width multiplier for the 3 different pretraining datasets: ImageNet, iNat21 and iNat21Mini. Results are shown in Fig. 7. MB3 and MB6 are the mobile inverted convolutions with expansion ratio of 3 and 6 respectively. The grey lines denote the downsampling of image due to strided convolutions. All architectures are followed by a pooling layer to reduce the image size to $1 \times 1$. In contrast to standard handcrafted architectures, larger $7 \times 7$ convolutions are preferred even in the later stages of the network. We also see that the majority of the preferred convolutions is MB6 $7 \times 7$ suggesting that the network prefers convolutions with more parameters for the self-supervised regime due to lots of data. This is less preferred in smaller datasets like iNat21Mini where MB3 convolutions are common especially in earlier stages of the network. It is difficult to draw conclusions on the type of network preferred between ImageNet and iNat21 showing that it is imperative to search for an optimal architecture rather than handcraft them.

## E    DOWNSTREAM DATASET PERFORMANCE CORRELATION WITH IMAGENET

We show the downstream dataset correlation for all 10 downstream datasets in addition to ImageNet-1K Deng et al. (2009): CIFAR10/100 Krizhevsky et al. (2009), Stanford Cars Krause et al. (2013) and Dogs Khosla et al. (2011), CUB-200 Welinder et al. (2010), MIT-67 Quattoni & Torralba (2009), SVHN Netzer et al. (2011), Flowers-102 Nilsback & Zisserman (2008), FGVC-Aircraft Maji et al. (2013), Sports8 Li & Fei-Fei (2007). These are shown for both ResNets (Fig. 8) and MobileNets (Fig. 9). We see similar results for the 5 datasets in addition to those shown in Fig. 3 of main paper. High correlation exists for datasets which are visually similar to ImageNet while it is less correlated for datasets which are out of domain. For MobileNets this correlation is even less pronounced with high variance in performance at higher ImageNet accuracies.

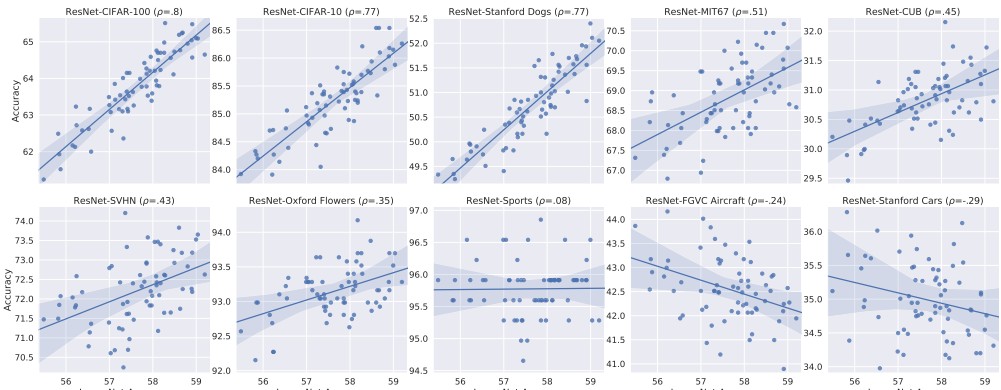

Figure 8: **ImageNet performance correlation with 10 different downstream datasets for various ResNets.** We show linear evaluation top-1 accuracy of ImageNet (x-axis) vs. different downstream datasets (y-axis) obtained from variations of ResNet; all models are pretrained on ImageNet-1K Deng et al. (2009) using SimCLR Chen et al. (2020a) under the same protocol. The solid lines and shaded areas indicate fitted regression models and their confidence intervals.

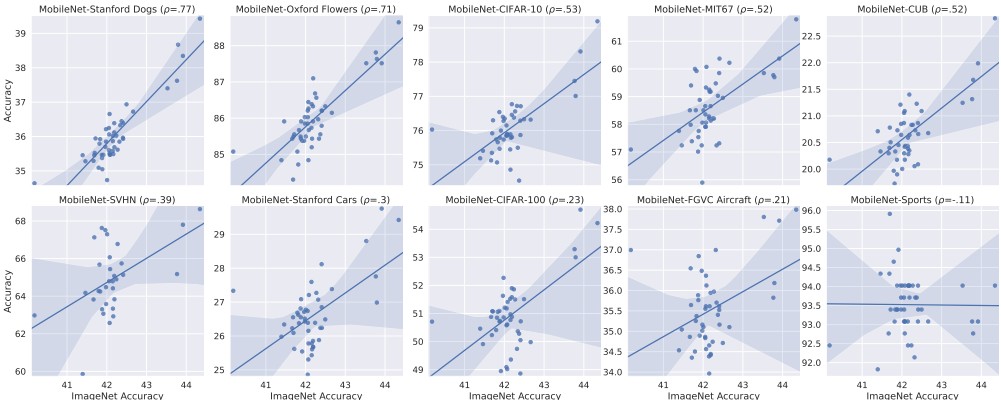

Figure 9: **ImageNet performance correlation with 10 different downstream datasets for various MobileNets.** We show linear evaluation top-1 accuracy of ImageNet (x-axis) vs. different downstream datasets (y-axis) obtained from variations of MobileNet; all models are pretrained on ImageNet-1K Deng et al. (2009) using SimCLR Chen et al. (2020a) under the same protocol. The solid lines and shaded areas indicate fitted regression models and their confidence intervals.

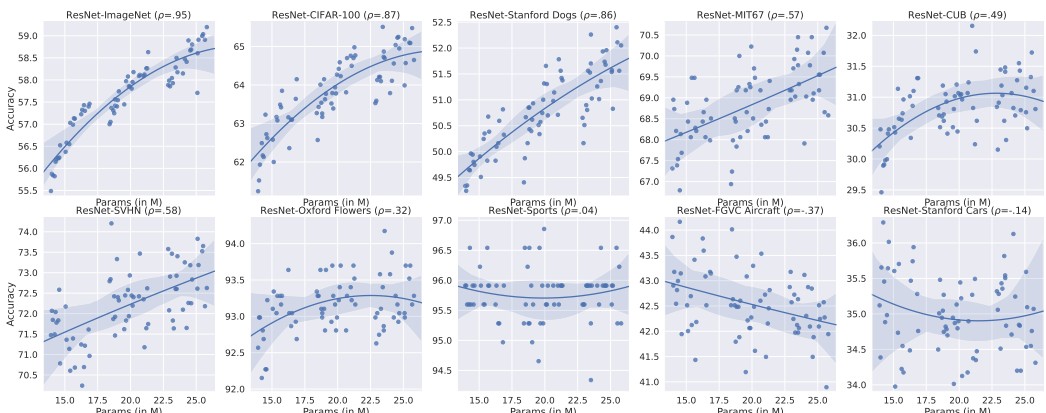

Figure 10: **Dataset performance correlation with 10 different datasets for various ResNets.** We show the model size in terms of parameter counts (x-axis) vs. top-1 accuracy on different datasets obtained from variations of ResNet-like architectures; all models are pretrained on ImageNet-1K Deng et al. (2009) using SimCLR Chen et al. (2020a) under the same protocol. The solid lines and shaded areas indicate fitted regression models and their confidence intervals.

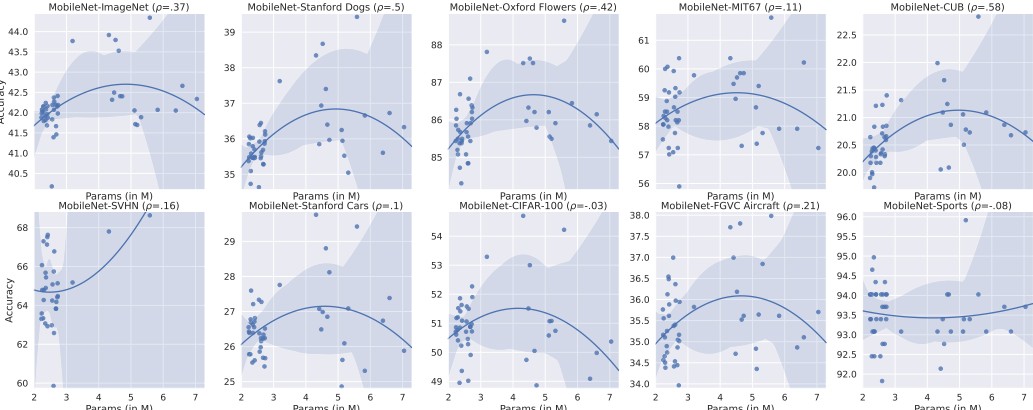

Figure 11: **Dataset performance correlation with 10 different datasets for various MobileNets.** We show the model size in terms of parameter counts (x-axis) vs. top-1 accuracy on different datasets obtained from variations of MobileNet-like architectures; all models are pretrained on ImageNet-1K Deng et al. (2009) using SimCLR Chen et al. (2020a) under the same protocol. The solid lines and shaded areas indicate fitted regression models and their confidence intervals.

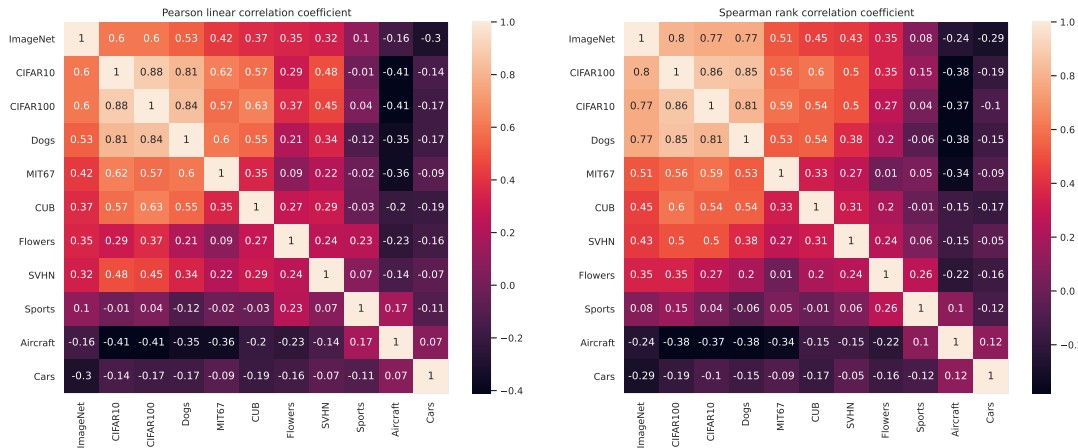

Figure 12: **Linear and rank correlation of ResNets between different pairs of 11 datasets** We show correlation between every pair of top-1 accuracy on 11 downstream tasks obtained from ImageNet-pretrained ResNets. We see no strong correlation except for ones highly similar to the data the models were originally pretrained on, e.g., CIFAR-10/100 and Dogs120.

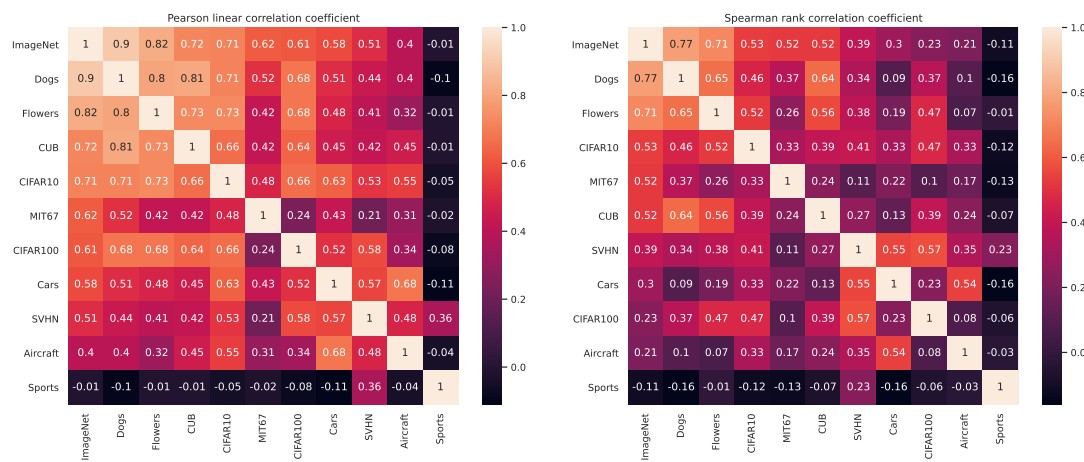

Figure 13: **Linear and rank correlation of MobileNets between different pairs of 11 datasets** We show correlation between every pair of top-1 accuracy on 11 downstream tasks obtained from ImageNet-pretrained MobileNets. The correlation

# F  DATASET PERFORMANCE AS A FUNCTION OF NUMBER OF PARAMETERS

We show the dataset correlation with respect to number of parameters for 5 more datasets in addition to that shown in Fig. 4 of main paper. Fig. 10 summarizes the results for ResNets while Fig. 11 shows results for MobileNets. We see that similar results hold for the additional 5 datasets where more parameters, and consequently larger networks, does not always lead to better downstream performance.

# G  LINEAR AND RANK CORRELATION FOR RESNETS/MOBILENETS

We show the summary of the correlation across different datasets for both ResNets (Fig. 12) and MobileNets (Fig. 13). In addition to Spearman's rank correlation coefficient, we also show Pearson's linear correlation coefficient. While Pearson's linear coefficient is higher in the case of MobileNets, the linear fit still exhibits high variance for higher ImageNet accuracies, as seen in Fig. 9.

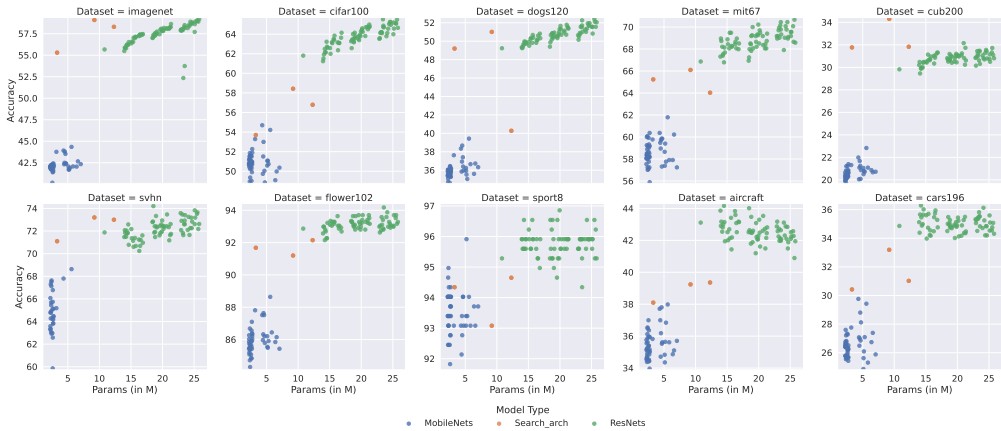

Figure 14: **ImageNet performance correlation with 10 different downstream datasets for various ResNets, MobileNets and searched architectures.** The searched architectures outperform MobileNets while being comparable with ResNets at fewer parameters.

## H  DATASET LICENSES

Table 5 lists some datasets we used and their licenses.

Table 5: **Licenses of datasets**.

| Dataset | License |
|---|---|
| CIFAR-10 Krizhevsky et al. (2009) | MIT |
| CIFAR-100 Krizhevsky et al. (2009) | MIT |
| ImageNet Deng et al. (2009) | BSD 3-Clause |
| Sport8 Li & Fei-Fei (2007) | CC0: Public Domain |
| Stanford Dogs Khosla et al. (2011) | BSD 3-Clause |
| Stanford Cars Krause et al. (2013) | BSD 3-Clause |
| CUB-200 Welinder et al. (2010) | Data files © Original Authors |
| MIT-67 Quattoni & Torralba (2009) | MIT |
| SVHN Netzer et al. (2011) | CC0: Public Domain |
| Flowers-102 Nilsback & Zisserman (2008) | GNU General Public License, version 2 |

