# OpenReview forum: "Moving Beyond Handcrafted Architectures in Self-Supervised Learning"
_ICLR.cc/2023/Conference — Submitted to ICLR 2023_

### Official Review · Reviewer_5FVM · 2022-10-24

**Confidence:** 4
**Correctness:** 4
**Technical Novelty And Significance:** 3
**Empirical Novelty And Significance:** 3
**Recommendation:** 6

**Clarity, Quality, Novelty And Reproducibility:**

Clarity: Good

Quality: Good

Novelty: Fair

Reproducibility: Fair

**Strength And Weaknesses:**

Pros:
1. The motivation is clear. I think it is important to rethink the design of architectures in the context of self-supervised learning.
2. This paper provides insight for future research. For instance, Fig. 3 shows that larger models do not always perform better in SSL, which is counterintuitive but interesting.

Cons:
1. Can the problems of Figures 1 to 3 be solved if the author's proposed method (NAS+SSL) is used? Is it true, for example, that the larger the size of the model discovered by NAS, the better the performance? Are the problems with the hand-designed models in Figure 2 still present? I have the same question for Figures 1&3 and I hope the authors can draw Figures 1 to 3 with the NAS searched model for comparison.

2. What can we conclude from the results of the NAS to design a model for SSL? Currently, SSL gradually surpasses supervised learning on various downstream tasks by pretraining weights. So I am curious if NAS under SSL can also outperform NAS under supervised learning, by pretraining the model structure.







**Summary Of The Paper:**

This paper investigates the role of architectures in self-supervised learning. The authors conduct exhaustive experiments with different architectures and downstream datasets under the setting of SSL. They show that there is no one network that performs consistently well across the scenarios and hence they propose to learn the network architectures in SSL. Experimental results show that self-supervised searched architectures outperform handcrafted ones across different downstream datasets.

**Summary Of The Review:**

I am willing to raise my score if the authors can address my concerns.

---

> ### Author Response · Authors · 2022-11-17
> **Author response**
>
> Thank you for reviewing the paper and providing your feedback. We are glad to see that you found the motivation clear and the paper insightful. We address the major concerns below:
>
> 1. We show that NAS architectures confirm our various observations from Section 3.
> + **Do larger NAS models perform better?** Figure 1 to 3 show that for a large variety of architectures, there is no single architecture which consistently outperforms on all datasets. Additionally, larger models do not always perform better on some datasets. For 2 searched architectures with different parameters, we observe that the larger model does not always perform better, as shown in the table below. NAS overcomes the problem of hand-designed models as it automatically adjusts the depth/width/convolutional size of the network and finds the optimal architecture for a given search dataset.
> | No. | Params | ImageNet | CIFAR-10 | CIFAR-100 | CUB-200 | NA-Birds | Aircraft |
> |-----|--------|----------|----------|-----------|---------|----------|----------|
> | 1   | 9.2    | 59.06    | 81.8     | 58.4      | 34.3    | 26.1     | 39.2     |
> | 2   | 12.3   | 58.2     | 80.5     | 56.8      | 31.8    | 24.9     | 39.4     |
> + **Are top-heavy architectures more optimal than bottom-heavy or vice-versa?** The approach of NAS+SSL identifies the optimal architecture for a given dataset. In addition to overall depth/width, it also automatically optimizes for the architecture to be top-heavy or bottom-heavy and allocate the right portion of parameters as shown in Fig. 7 of the supplementary, where different stages of the network have different types of convolutional operations. Thus, it automatically solves the problem of choosing the right allocation of parameters for an architecture on a given dataset, without hand-tuning.
> + **Is ImageNet performance indicative of downstream performance?** We see that from Table 3 that ImageNet performance (or even iNat21 performance) shows similar correlation with in-distribution downstream datasets. This is apparent as searched architectures outperforms ResNet-50 on most in-distribution datasets. The problem of network generalization to out-of-distribution dataset persists as NAS searched architectures strongly encode the inductive biases of the dataset they are trained on.
>
> + **Comparison of performance of searched architecture with ResNets and MobileNets:** We show performance on the different datasets with respect to number of parameters of the network for the 3 cases of ResNets, MobileNets and the searched architecture in Fig. 14 of the appendix. We see that the searched architectures (orange) have much fewer parameters than ResNets (green) and far outperform MobileNets (blue) even in a similar parameter range (as also highlighted in Table 3).
>
> 2. **Conclusion of NAS for SSL:**
>
> + The main conclusion of NAS for SSL is that searched architectures outperform handcrafted ones on the dataset they are trained on. Most handcrafted architectures are suited to object-centric datasets like ImageNet and are unlikely to generalize to datasets with a large domain shift.  Additionally, we obtain much better performance with NAS architectures compared to baseline handcrafted networks like MobileNet-V2 (~13% on ImageNet with similar parameters). This is a bigger gap than in supervised training setups where searched architectures [1] obtain a ~3% higher performance on ImageNet compared to the baseline MobileNet-V2.
>
> + While SSL methods are approaching the performance of supervised performance on downstream tasks, SimCLR [2] in particular obtains lower performance on linear evaluation when compared with supervised pretrained weights. Since our work applies SimCLR to NAS, it is likely that an architecture searched using supervised learning outperforms that of SimCLR. The architecture would utilize label information during the search phase which is crucial to the linear evaluation metric and hence is not a fair comparison. Nevertheless, we are currently working on obtaining results for NAS searched under supervised learning but evaluated on SSL and will provide them as permitted by time and computational constraints.
>
>
> [1] Cai, Han, Ligeng Zhu, and Song Han. "Proxylessnas: Direct neural architecture search on target task and hardware." arXiv preprint arXiv:1812.00332 (2018).
>
> [2] Chen, Ting, et al. "A simple framework for contrastive learning of visual representations." International conference on machine learning. PMLR, 2020.

---

### Official Review · Reviewer_JRjb · 2022-10-26

**Confidence:** 5
**Correctness:** 2
**Technical Novelty And Significance:** 2
**Empirical Novelty And Significance:** 2
**Recommendation:** 3

**Clarity, Quality, Novelty And Reproducibility:**

- Good clarity and reproducibility.
- Please refer to the previous section for possible problems in quality and novelty.

**Strength And Weaknesses:**

Strength
- Extensive experiments and analysis.
- Interesting perspective and attempt to try NAS in self-supervised learning.

Weaknesses
- In short, I think this paper does not decouple the advantages of NAS itself from SSL, while the architecture obtained from the search may have significant advantages in terms of MAdds.

-----

- **About the statements in Section 3** I agree with this statement "ImageNet performance is not indicative of downstream performance for SSL" alone, as evidenced by many previous attempts in pixel-level contrastive learning like DenseCL and PixPro. However, the statement in here does not support the main point of this paper.
	- First of all, the paper does not give a specific setting for the 116 sampled models, which means that many of them may be in a situation where they perform poorly and are unlikely to be adopted.
	- In addition, the sampling of the MobileNet structure is too concentrated in terms of the size and performance of the sampled models, as illustrated by the distribution of ImageNet results for MobileNet in Figure.2, the model size for MobileNet in Figure.3, and the top/bottom param ratio in Figure.4.
	- And this means that the conclusion obtained using spearman rank correlation on such a small space has a high probability of being problematic. We can also see from Figure.13 in the Appendix that MobileNet presents better correlations than ResNet on almost most of the datasets when Pearson coefficients are used. For the few remaining datasets, the lower overlap (e.g. Aircraft) or higher performance (e.g. Sports 8, also too small with only about 1.5K images in total) do not make it universally available.
	- At the same time, if the analysis is performed only at the architectural level, the influence of SSL on it should be stripped, because this part of SSL are pre-trained on ImageNet-1K while using linear evaluation as a metric, which will inevitably be helpful for ImageNet-1K related tasks (e.g. CIFAR). If measured using supervised training, I think the performance on ImageNet would be at least highly correlated with the performance of other **classification tasks** (Especially considering the higher correlation already seen in terms of Pearson coefficients so far).

- The experimental section is missing an important baseline, i.e., the performance of the model obtained by using supervised training instead of SSL for search.
- For MobileNet, MAdds is a very important metric. The model given in the paper seems to have an overall stride of only 16 (inferred from Appendix D), and the latency loss is removed in the search process, which may result in a model with significantly larger MAdds than MobileNetV2.


**Summary Of The Paper:**

This paper attempts to combine the neural architecture search (NAS) process with self-supervised learning (SSL) so as to learn structures suitable for pre-trained datasets (e.g. ImageNet-1K, iNat-2021) in a self-supervised task like SimCLR. Extensive analysis and experiments are included to demonstrate its arguments.

**Summary Of The Review:**

This paper wants to show that for SSL, the architecture obtained by NAS will have a significant advantage over the handcrafted architecture. However, neither the theoretical analysis nor the experimental part can exclude the benefits from the NAS itself, and no further experiments have been conducted for other self-supervised learning methods.

---

> ### Author Response · Authors · 2022-11-17
> **Author response**
>
> We thank the reviewer for their valuable feedback. We address the concerns below:
>
> **1) ImageNet performance is not indicative of downstream performance for SSL:**
>
> We agree with the reviewer that the works of DenseCL and PixPro show that ImageNet performance is not correlated with downstream performance. However, they show that the contrastive learning objectives are not optimized to *dense prediction problems like semantic/instance segmentation* while also using a fixed ResNet-50 backbone. Our observations are on *instance level classification problem* while focusing on an orthogonal direction for measuring correlation: varying the network architectures for a fixed objective.
>
> **2) Sampled models may perform poorly:**
>
> We instantiate the networks based on number of blocks at each stage of the network as well as the width. We thank the reviewer for their suggestion and a more detailed explanation of the sampling is provided in the appendix in Section C. As observed from the top-left subplot in Figure 3, many of the sampled networks lie on the pareto-frontier curve and hence do not perform poorly. They achieve similar ImageNet performance as the heavily optimized ResNet-50 at the same parameter range.
>
> **3) MobileNets are sampled very closely and have high Pearson’s linear correlation coefficient:**
>
> We agree that a majority of the MobileNets are sampled closely and lie in the 2-3M parameter range. However, beyond this range, there is a wide swing in network performance, as seen in the bottom plots of Fig. 3. Even within this 2-3M range, we see swings in network performance for different datasets like CIFAR-100/Stanford Cars/Aircraft (~3%). We agree that Spearman’s rank correlation coefficient can be noisy in a small space. However, we observe from the figures itself, there is little correlation between ImageNet and downstream performance or between downstream performance and number of parameters.
>
> Pearson’s correlation coefficient in particular is a harmful metric to measure correlation in this space as there are no observable linear trends due to large error intervals of the linear fit (Fig. 9). The coefficients themselves have a large confidence interval and should not be directly used to judge the correlation between various datasets. They were included for the sake of completeness.
>
> **4) ImageNet-downstream dataset correlation should be measured using supervised training:**
>
> We agree that supervised ImageNet performance is likely to be better correlated with downstream tasks as also shown by [1]. However, the main objective of our approach is to show the benefit of *harnessing NAS in aid of SSL* and not for supervised learning, as highlighted in the beginning of Section 4. Many prior works have applied NAS learnt without labels [2,3,4] and applied to supervised learning which is not the target of our approach.
>
> **5) Networks searched using supervised objectives should be used as a baseline:**
>
> A network searched using supervised learning utilizes label information to learn the optimal architecture. Such a network encodes stronger dataset biases and is likely to perform better when evaluated using the linear classification metric and would therefore not be a fair comparison to a network searched using the SimClr objective (which performs far worse than its supervised counterpart especially with smaller batch size and training epochs). Nevertheless, we are working on providing an experiment to provide a comparison of a supervised architecture vs a self-supervised architecture for SSL, permitting time and computational constraints.
>
> **6) Searched architectures may have significantly higher MAdds:**
>
> The overall stride of the network is 32 and not 16 as the first convolution downsamples the image by a factor of 2 as well (similar to [5]), as mentioned in text in Section C. Due to this, it does not have significantly higher MAdds. We thank the reviewer for pointing it out and will update the figure to reflect that. We provide the number of MAdds in the table below and see that the network does not have a significantly higher number of MAdds while significantly outperforming the baseline MobileNet. Additionally, the number of MAdds is lower than that of ResNet-18 while still maintaining higher performance.
>
> | Architecture  | MACs (in B) | Params (in M) | Top-1 ImageNet Acc. (%) |
> |---------------|-------------|---------------|-------------------------|
> | MobileNet-V2  | 0.33        | 3.5           | 41.9                    |
> | ResNet-18     | 1.82        | 11.69         | 49.8                    |
> | Searched Arch | 0.56        | 3.34          | 55.3                    |

---

> > ### Author Response · Authors · 2022-11-17
> > **References**
> >
> > [1] Kornblith, Simon, Jonathon Shlens, and Quoc V. Le. "Do better imagenet models transfer better?." Proceedings of the IEEE/CVF conference on computer vision and pattern recognition. 2019.
> >
> > [2] Liu, Chenxi, et al. "Are labels necessary for neural architecture search?." European Conference on Computer Vision. Springer, Cham, 2020.
> >
> > [3] Yan, Shen, et al. "Does unsupervised architecture representation learning help neural architecture search?." Advances in Neural Information Processing Systems 33 (2020): 12486-12498.
> >
> > [4] Li, Changlin, et al. "Bossnas: Exploring hybrid cnn-transformers with block-wisely self-supervised neural architecture search." Proceedings of the IEEE/CVF International Conference on Computer Vision. 2021.
> >
> > [5] Cai, Han, Ligeng Zhu, and Song Han. "Proxylessnas: Direct neural architecture search on target task and hardware." arXiv preprint arXiv:1812.00332 (2018).

---

### Official Review · Reviewer_6dkS · 2022-11-07

**Confidence:** 4
**Correctness:** 3
**Technical Novelty And Significance:** 2
**Empirical Novelty And Significance:** 2
**Recommendation:** 6

**Clarity, Quality, Novelty And Reproducibility:**

[Clarity]
1. Clarified the assumptions of the current research.
2. Clarified the main objective of the study is to show that the choice of network is highly impactful in SSL and handcrafting the architecture is very hard.
3. Authors raised research questions in a clear context.
4. Clearly states the three main contributions
5. Clearly explains the reason behind the choice of the NAS algorithm and dataset


[Quality]
1. Appropriate references were made.
2. Tables, figures, and appendix are effectively supporting the arguments made.


[Novelty]
This work possesses novelty to a certain degree in that the author(s) tried to use NAS in aid of SSL, and conducted a large scale experiments on the variant of networks. However, the novelty is not substantial (limited) as there are a number of work that have similar approach in combining NAS and SSL. A preprint titled ‘CSNAS: Contrastive Self-supervised Learning Neural Architecture Search via Sequential Model-Based Optimization’ (https://arxiv.org/abs/2102.10557) is one of the examples. Despite the fact, as author(s) mentioned, in the beginning of Section 4, there is a distinction between their idea and prior work.


[Reproducibility]
1. Provided the details of 1) how they varied the architecture of each network they used , 2) the number of epochs and batch size when pretraining with the machine they used for training.
2. Provided the metric used (Spearman’s rank correlation coefficient) when evaluating the correlation between ImageNet and downstream performance.

**Strength And Weaknesses:**

[Strengths]

S1: The study found out the implicit underlying assumption of the current literature and pointed out that it can be incorrect.

S2: The results have shown that the SSL architecture outperforms the handcrafted architectures, and have included concrete experiments regarding the distributional shift.

[Weaknesses]

W1: Only one optimization objective (SimCLR) has been used. They could have checked if the proposing method works across many different learning objectives.

W2: Idea of combing NAS and SSL is not novel.

W3: In the downstream transfer experiment, the searched architecture did not show promising performance on out-of-distribution dataset, except for the comparison of MobileNetV2, and this results are not surprising.

W4: Although 116 variations of network were experimented, there are just two backbone network architectures.

**Summary Of The Paper:**

The study provides evidence that a network architecture plays a significant role in contrastive SSL, by utilizing 116 variants of ResNet and MobileNet architecture, which were evaluated across 11 downstream tasks in the contrastive SSL setting.
It showed that no one architecture demonstrated a consistently good result, thus suggesting future researchers focus on learning architecture as well as the weights of the network in the SSL setting. They conducted two experiments: 1) network variation experiment on downstream tasks with observation of correlation between the models in downstream performance, and 2) applying NAS algorithm to the SSL setting to search for the optimal architectures on unlabeled pretraining dataset via contrastive learning.


**Summary Of The Review:**

From the extensive study, readers can agree to the statement that there is no one architecture that performs well across different downstream tasks in SSL, and that NAS+SSL can be a solution for mitigating the problem, where researchers have to handcraft the network architecture. However, as mentioned in the section above, the novelty of this idea is limited in that the combination of NAS and SSL has been experimented widely in the literature.
However, this study opens up many possible research questions regarding the effectiveness on architecture search for SSL, thus can be act as a ground research of future studies.

---

> ### Author Response · Authors · 2022-11-16
> **Author response**
>
> We thank the reviewer for their feedback and providing constructive comments. We address the concerns below:
>
>
> **Different self-supervised learning objectives:**
>
> We agree that it would be interesting to evaluate the role of architectures across different contrastive SSL objectives. However, due to computational constraints of applying NAS to each of these approaches, we limit to SimClr due to its simplicity. A future study on whether the same architectures work across different SSL objectives would answer several questions but we restrict the scope of this paper to answer the question of whether architectures have an impact for some self-supervised learning objective.
>
> **Novelty of applying NAS to SSL:**
>
> We agree that several prior works exist, which focus on using SSL for NAS but apply it to supervised learning instead. However, we propose to *use NAS to improve upon SSL*. Our main goal of applying NAS is to simply show its effectiveness for SSL (which has previously not been looked at) and not for improving upon existing NAS algorithms. While we utilize existing NAS approaches, we believe that the key observations drawn from such works are also necessary for the community, similar to [1,2,3]. Our hope is that this work lays the groundwork for future research into the role of architectures in SSL.
>
> **Out-of-distribution performance is not very surprising:**
>
> While the results for out-of-distribution performance is not very surprising, it further shows that *NAS architectures adapt to the inductive biases specific to the dataset* they are searched on. Handcrafted architectures such as ResNet-50 might be heavily tuned towards ImageNet-like datasets and perform worse on datasets with significant domain shift unlike NAS which is highly flexible and can adapt to new data domains (as shown with the performance for iNat21 in Table 1)
>
> **Experiments are limited to two backbone architectures:**
>
> We provide experiments on ResNets and MobileNets because the former is the defacto architecture used for various self-supervised learning approaches while the latter is chosen mainly due to its lightweight nature and widespread application in low-resource setups. While there exist other architectural backbones such as DenseNets, EfficientNets or ShuffleNets and so on, and provide improvements over the 2 chosen baselines, they serve similar aspects to the two chosen backbones. MobileNets were preferred additionally because their insights served as a useful pointer to utilize ProxylessNAS, which has a MobileNet search space, as the chosen algorithm.
> We will update the main paper to better reflect these points.
>
>
> [1] Kolesnikov, Alexander, Xiaohua Zhai, and Lucas Beyer. "Revisiting self-supervised visual representation learning." Proceedings of the IEEE/CVF conference on computer vision and pattern recognition. 2019.
>
> [2] Cole, Elijah, et al. "When does contrastive visual representation learning work?." Proceedings of the IEEE/CVF Conference on Computer Vision and Pattern Recognition. 2022.
>
> [3] Gwilliam, Matthew, and Abhinav Shrivastava. "Beyond Supervised vs. Unsupervised: Representative Benchmarking and Analysis of Image Representation Learning." Proceedings of the IEEE/CVF Conference on Computer Vision and Pattern Recognition. 2022.

---

### Official Review · Reviewer_ZDfX · 2022-11-08

**Confidence:** 3
**Correctness:** 3
**Technical Novelty And Significance:** 1
**Empirical Novelty And Significance:** 1
**Recommendation:** 3

**Clarity, Quality, Novelty And Reproducibility:**

**Clarity**:
The discussions of the experiment setting and results are sufficient. However, an introduction to the SSL and NAS is required to make the paper self-contained.

**Quality**:
The experiment section is qualified but


**Novelty**:
There is no new model or algorithm provided in the paper.  The heuristic conclusion from the empirical study is not surprising and lacks of scientific rigors. The novelty of applying NAS in SSL is limited.

**Reproducibility**:
No code is provided for this empirical study, so the difficulty of reproducing the results is high.

**Strength And Weaknesses:**

**Strength**: the structure of the paper is clear and the paper is well-written. The experiments are sufficient and well-designed.


**Weaknesses**:
The novelty of the paper is very limited. There are several messages from section 3:

    1. ImageNet performance is not indicative of downstream performance for SSL.
    2. Larger networks do not always perform better in contrastive SSL.
    3. There is no winner in the battle of top vs. bottom heavy networks in SSL.
    4. It is necessary to allocate the right portion of parameters to different layers of a given network topology.

All the messages are trivial,  not mathematically rigorous, and are generally observed in different machine-learning tasks.    Although I do agree that the empirical study is sufficient  as a motivation and can support the  general key takeaway: ``we need
to move beyond handcrafted architectures in SSL'', but section 3 doesn't convey any new insights about SSL.

The second part of the paper (section 4) applies NAS in SSL, this combination is also a trivial extension and the resulting improvement is not surprising.

I suggest the author treat the empirical study as the first step towards designing a new NAS algorithm to improve the SSL results or provide a more theoretical understanding of the described phenomenon.


**Others**:
1. The title of table 1 is misleading,  ``NAS vs handcraft architecture '' will be more clear.




**Summary Of The Paper:**

This paper conducts an empirical investigation on the role of architectures in self-supervised learning. Based on the empirical study, the authors apply the neural architecture search in the current SSL framework and demonstrate improved performance in several downstream tasks.

**Summary Of The Review:**

I recognize the value of empirical study and believe the architecture choice is crucial for semi-supervised learning.
However, the novelty of the paper is limited and the message of the study is vague, unsurprising, and not rigorous. The paper will be more suitable for venues like workshops or benchmark tracks.

---

> ### Author Response · Authors · 2022-11-16
> **Author response**
>
> Reviewer ZDfX
>
> Thank you for reviewing the paper and providing your feedback. We respectfully disagree with the reviewer on the triviality of applying NAS to SSL and address the major points below:
>
>
> ***1) Messages are trivial***
>
> The main aim of our paper is to draw attention to the role of architectures for contrastive self-supervised learning. Many prior works exist which focus on other aspects of SSL such as loss objectives [1,2], pretraining datasets [3] or evaluation benchmarks [4], but are *limited to the fixed ResNet-50 architecture*. The **effect of architectures is largely overlooked.**
>
> Our findings from section 3 draw key insights about the trends observed for different aspects of a handcrafted network. While similar results may hold true in other learning setups such as supervised learning, it is by no means trivial and is actually **harmful to draw the same conclusions in the contrastive SSL regime** as highlighted by [5] (‘’*We challenge a number of common practices in self supervised visual representation learning and observe that standard recipes for CNN design do not always translate to self-supervised representation learning*.’’).
>
> The results from Table 1 are *not* trivial and highly encouraging as the searched architectures slightly outperform the handcrafted ResNet-50 on ImageNet with fewer parameters. It shows an even bigger gain for the dataset of iNat21 (+2.5% top-1 acc.) showing that the **searched architectures better adapt to the training dataset**. This is in contrast to supervised learning, where the performance of the searched architecture on ImageNet itself (75.1%) is lower than ResNet-50 performance (77.2%) [6].
>
> Our work is the first to provide extensive empirical evidence to support our observation that architecture matters for contrastive self-supervised learning. Moreover, we show that **NAS is crucial for datasets with a significant domain shift from object-centric datasets**, which is an important observation to the community which conventionally uses hand-crafted networks like ResNet-50 as the backbone for a variety of datasets and tasks.
>
>
> ***2) Novelty in application to SSL***
>
> The focus of application of NAS to SSL is to show that we can *harness the benefits of NAS for contrastive SSL as well and not to outperform the state-of-the-art in NAS* as highlighted in the beginning of section 4. Our hope is to draw attention to the effectiveness of NAS towards SSL and lay the foundation for future research work.
>
> Our key insights from the empirical study in section 3 indeed drive our choice of using ProxylessNAS as the preferred NAS approach. Its flexibility to choose different convolutional operations at different stages of the network allows for learning larger or smaller networks, and allocating the preferred portion of parameters to different layers in the network. It is efficient and scalable, crucial for SSL setups which typically have high memory and compute requirements.
>
> We thank the reviewer for the suggestion and will update the title of Table 1 to ‘’Searched architectures vs hand-crafted architectures’’ to make it clearer.
>
> [1] Chen, Ting, et al. "A simple framework for contrastive learning of visual representations." International conference on machine learning. PMLR, 2020.
>
> [2] Caron, Mathilde, et al. "Unsupervised learning of visual features by contrasting cluster assignments." Advances in Neural Information Processing Systems 33 (2020): 9912-9924.
>
> [3] Cole, Elijah, et al. "When does contrastive visual representation learning work?." Proceedings of the IEEE/CVF Conference on Computer Vision and Pattern Recognition. 2022.
>
> [4] Gwilliam, Matthew, and Abhinav Shrivastava. "Beyond Supervised vs. Unsupervised: Representative Benchmarking and Analysis of Image Representation Learning." Proceedings of the IEEE/CVF Conference on Computer Vision and Pattern Recognition. 2022.
>
> [5] Kolesnikov, Alexander, Xiaohua Zhai, and Lucas Beyer. "Revisiting self-supervised visual representation learning." Proceedings of the IEEE/CVF conference on computer vision and pattern recognition. 2019.
>
> [6] He, Kaiming, et al. "Deep residual learning for image recognition." Proceedings of the IEEE conference on computer vision and pattern recognition. 2016.

---

### Author Response · Authors · 2022-12-11
**Author response**

Dear reviewers,

We thank you for your time and providing feedback for our work. We are pleased to see that you found the paper clear and well written (ZDfX) with extensive (JRjb) and concrete (6dkS) experiments, providing insight for future research (5FVM).

One of the main concerns raised is regarding the novelty of our approach. We would like to reiterate our point that the main aim of our work is not to develop a new state-of-the-art but to highlight the importance of using NAS in aid of SSL (which has been largely overlooked by the community) and lay the foundation for more works in this field, as also pointed out by reviewer 5FVM ( ‘’This paper provides insight for future research’’)  and reviewer 6dkS (‘’act as a ground research of future studies’’). Such studies with crucial insights [1,2,3,4] are also important for the community and should not be dismissed citing lack of novelty.

We have provided detailed responses to each reviewer for the individual concerns raised. With ICLR being a discussion-encouraging conference, we sincerely request the reviewers to communicate any further comments if any of their concerns remain unresolved after our responses. We hope that you can improve the rating otherwise, if your queries are addressed.

[1] Kornblith, Simon, Jonathon Shlens, and Quoc V. Le. "Do better imagenet models transfer better?." Proceedings of the IEEE/CVF conference on computer vision and pattern recognition. 2019.

[2] Cole, Elijah, et al. "When does contrastive visual representation learning work?." Proceedings of the IEEE/CVF Conference on Computer Vision and Pattern Recognition. 2022.

[3] Kolesnikov, Alexander, Xiaohua Zhai, and Lucas Beyer. "Revisiting self-supervised visual representation learning." Proceedings of the IEEE/CVF conference on computer vision and pattern recognition. 2019.

[4] Gwilliam, Matthew, and Abhinav Shrivastava. "Beyond Supervised vs. Unsupervised: Representative Benchmarking and Analysis of Image Representation Learning." Proceedings of the IEEE/CVF Conference on Computer Vision and Pattern Recognition. 2022.

---

### Decision · Program_Chairs · 2023-01-20

**Decision:**

Reject

**Justification For Why Not Higher Score:**

There isn't enough enthusiasm amongst the reviewers (nor myself) to support this paper to be accepted in its current form. It's essentially an interesting observation (which also isn't particularly surprising).

**Justification For Why Not Lower Score:**

N/A

**Metareview: Summary, Strengths And Weaknesses:**

The paper discusses the role of architecture in contrastive SSL, finding that there is no clear recommended architecture that provides good downstream task performance. The authors therefore argue to search for an appropriate architecture. Whilst the authors back us this statement with experimental evidence, the main issue the reviewers had was how significant this observation is. There is no one willing to champion this paper and I think the consensus is that whilst this is an interesting piece of work, it doesn't rank highly enough for ICLR.